# Induced Pluripotent Stem Cells: Hope in the Treatment of Diseases, including Muscular Dystrophies

**DOI:** 10.3390/ijms21155467

**Published:** 2020-07-30

**Authors:** Daniela Gois Beghini, Samuel Iwao Horita, Cynthia Machado Cascabulho, Luiz Anastácio Alves, Andrea Henriques-Pons

**Affiliations:** Laboratório de Inovações em Terapias, Ensino e Bioprodutos, 2- Laboratório de Comunicação Celular, both at the Instituto Oswaldo Cruz, Fundação Oswaldo Cruz, Rio de Janeiro RJ 21040-900, Brazil; samuel.horita@ioc.fiocruz.br (S.I.H.); cynthiac@ioc.fiocruz.br (C.M.C.); alveslaa@ioc.fiocruz.br (L.A.A.); andreah@ioc.fiocruz.br (A.H.-P.)

**Keywords:** induced pluripotent stem cells, regeneration, cellular therapy, stem cells, muscular dystrophy

## Abstract

Induced pluripotent stem (iPS) cells are laboratory-produced cells that combine the biological advantages of somatic adult and stem cells for cell-based therapy. The reprogramming of cells, such as fibroblasts, to an embryonic stem cell-like state is done by the ectopic expression of transcription factors responsible for generating embryonic stem cell properties. These primary factors are octamer-binding transcription factor 4 (Oct3/4), sex-determining region Y-box 2 (Sox2), Krüppel-like factor 4 (Klf4), and the proto-oncogene protein homolog of avian myelocytomatosis (c-Myc). The somatic cells can be easily obtained from the patient who will be subjected to cellular therapy and be reprogrammed to acquire the necessary high plasticity of embryonic stem cells. These cells have no ethical limitations involved, as in the case of embryonic stem cells, and display minimal immunological rejection risks after transplant. Currently, several clinical trials are in progress, most of them in phase I or II. Still, some inherent risks, such as chromosomal instability, insertional tumors, and teratoma formation, must be overcome to reach full clinical translation. However, with the clinical trials and extensive basic research studying the biology of these cells, a promising future for human cell-based therapies using iPS cells seems to be increasingly clear and close.

## 1. Induced Pluripotent Stem Cells–General Concepts

Stem cells can be classified as totipotent, pluripotent, or multipotent cells according to their biological source and the capacity to differentiate into other cell types. Totipotent stem cells are found very early in embryonal development and can differentiate into all cell types in the organism, as well as into extraembryonic tissues. Pluripotent cells can be isolated from blastocysts or the umbilical cord immediately after birth, and are also able to differentiate into all tissue cells, except extraembryonic structures. However, certain disadvantages must be observed when considering these stem cells in regenerative medicine. These include the high risk of rejection and ethical issues when the isolation is performed from embryos. On the other hand, due to their high plasticity, pluripotent stem cells are considered ideal to obtaining the multiple cell types required after stem cell-based therapies.

Multipotent stem cells are isolated from adult tissues and have no ethical issues involved. These include hematopoietic, mesenchymal, and neural stem cells. Multipotent stem cells can be isolated from the patients subjected to treatment, with no risk of rejection, and be expanded in vitro for transplant. However, these cells display reduced plasticity, as they can only differentiate into specialized cell types present in specific tissues or organs, their main disadvantage. The ideal cellular population best suited for stem cell-based therapies should combine the high plasticity of embryonic stem cells and the convenient isolation from patients under treatment. To this end, induced pluripotent stem (iPS) cells were generated using embryonic or adult somatic cells. The somatic cells are subjected to the ectopic expression of transcription factors that induce the stem cell-like properties and the high plasticity required for cell therapy. Therefore, iPS cells can potentially revolutionize the field of regenerative medicine and provide new tools for stem cell research.

In the nineties, it was demonstrated that somatic cells could be reprogrammed to an undifferentiated state by transferring their nuclear content into unfertilized oocytes [1]. These results showed that cellular differentiation is reversible. Later, the resetting of a somatic epigenotype to a totipotent state was successfully achieved when adult thymocytes were fused with embryonic stem cells [2]. These and other pioneering studies [3] paved the way for the Nobel prize-awarded paper published by Takahashi and Yamanaka [4], who hypothesized that “…factors that play important roles in the maintenance of embryonic stem cell identity also play pivotal roles in the induction of pluripotency in somatic cells”. In this study, mouse embryonic and adult fibroblasts were genetically reprogrammed to a pluripotent state, and the authors coined the term “iPS cells”. These cells were generated by using a retrovirus-based gene transfer system carrying the octamer-binding transcription factor 4 (Oct3/4), sex determining region Y-box 2 (Sox2), Krüppel-like factor 4 (Klf4), and c-Myc transcription factors, all involved in pluripotency maintenance in embryonic stem cells [4].

IPS cell technology brings great promise to medicine, such as personalized cell therapy, disease modeling, and new drug development and screening. However, some challenges must be circumvented, such as reprogramming efficiency and the risks associated with chromosomal instability, insertional tumor development, and teratoma formation. In this context, here, we review the literature, present the main methods of cell reprogramming, and show some initial results of clinical tests. Besides, we discuss the possibility of applying iPS cells in the treatment of muscular dystrophies. 

## 2. What Are the Main Methods to Reprogram Somatic Cells into iPS Cells?

Various delivery methods have been used to insert reprogramming factors into somatic cells. These approaches can be divided into integrative, which involves the insertion of exogenous genetic material into the host genome, and non-integrative methods. The integrative systems include the use of viral vectors (lentivirus, retrovirus, and inducible or excisable retro or lentivirus) and non-viral vectors (linear or plasmid DNA fragments and transposons). Likewise, non-integrative systems include viral (Sendai virus and adenovirus) and non-viral vectors (episomal DNA vectors, RNAs, human artificial chromosomes (HAC), proteins, and small molecule compounds) [5,6] (Figure 1 and Table 1).

The expression of primarily just four transcription factors (c-Myc, Klf4, Oct4, and Sox2) is sufficient to reprogram somatic cells into a pluripotent state. The discovery of those factors related to the embryonic stem cell phenotype allowed the production of embryonic stem-like cells first from mouse embryonic and adult murine fibroblasts [4] and then from adult human dermal fibroblasts [7,8]. The Oct4 seems to be the most important reprogramming factor, whereas Klf4 and c-Myc can be replaced by Nanog and Lin28, for example [9].

The first experiments achieved the conversion of somatic cells into iPS cells using retroviral or lentiviral transduction of the transcription factors. However, these vectors become integrated into the cell genome and represent a risk of insertional mutagenesis [10]. Moreover, they may leave residual transgene sequences as part of the host genome, leading to unpredictable alterations in the phenotype of downstream applications. To reduce multiple proviral integrations of the transcription factors and to increase the efficiency of the retrovirus- or lentivirus-based reprogramming process, polycistronic RNA viral vectors were created. These constructs allowed the expression of all reprogramming factors driven by a single promoter, reducing the number of genomic insertions [11]. Once the integration of the reprogramming factors is achieved, it is also essential to control the extent of expression. To this end, the use of excisable Cre-loxP technology for site-specific recombination and inducible tetracycline- or doxycycline-based vector systems allowed greater control of inserted genes expression, reducing inefficient silencing and uncontrolled reactivation [5].

It is important to highlight that other factors have been described as being able to induce cellular pluripotency and self-renewal. Besides, several types of somatic cells have also been subjected to in vitro reprogramming, such as pancreatic β cells, neural stem cells, stomach and liver cells, mature B lymphocytes, melanocytes, adipose stem cells, and keratinocytes. These results are summarized in the review published by Oldole and Fakoya [5].

The integrative non-viral technologies used to obtain iPS cells are based on the transference of DNA sequences using liposomes or electroporation [5], for example. It was possible to reprogram both mouse and human fibroblasts using a single multiprotein expression vector comprising the coding sequences of c-Myc, Klf4, Oct4, and Sox2 linked with 2A peptide [24]. When this single vector-based reprogramming system was combined with a piggyBac transposon, the authors successfully established reprogrammed human cell lines from embryonic fibroblasts with sustained pluripotency markers expression. PiggyBac is a mobile genetic element that includes a transposase enzyme that mediates gene transfer by targeted insertion and excision in the DNA. Moreover, Woltjen and collaborators showed the efficient reprogramming of murine and human embryonic fibroblasts using doxycycline-inducible transcription factors delivered by PiggyBac transposition. The authors also showed that the individual PiggyBac insertions could be removed from the iPS cell lines [15], being completely excised from its integration site in the original DNA sequence [25], which is a significant advantage.

The integrative methods for random or site-specific DNA insertion can affect normal cell function and physiology, including the transformation for tumorigenic cells, proliferation, and apoptosis control. Therefore, non-integrating viral vectors were constructed to generate iPS cells, the most promising of which is the Sendai virus, a negative-strand RNA virus [26]. The Sendai virus has the advantage of being an RNA virus that does not enter the nucleus and can produce large amounts of proteins [27]. Adenoviruses are also non-integrating viruses that appear to be excellent expression vehicles to generate iPS cells. They show DNA demethylation (a characteristic of reprogrammed cells), express endogenous pluripotency genes, and can generate multiple cells and tissues. However, the reprogramming efficiency of adenoviral vectors is only 0.001%–0.0001% in mouse [28] and 0.0002% in human cells [29], several orders of magnitude lower, when compared to lentiviruses or retroviruses [5]. The use of viruses, even in non-integrating systems, requires refined steps to exclude reprogrammed cells with active replicating viruses. Moreover, viral vectors may elicit an innate and adaptive immune response against viral antigens after the transplant to patients. In this case, the transplanted cells would become the target of molecular and cellular cytotoxic pathways, directly compromising the engraftment and therapy success.

Non-integrating non-viral systems include the transient expression of reprogramming factors inserted as combined episomal minicircles or plasmids. These contain the complementary DNA (cDNA) of Oct3/4, Sox2, and Klf4 and another plasmid containing the c-Myc cDNA, for example. This technique resulted in iPS cells with no evidence of plasmid integration [16], suggesting that episomal plasmids may be the best option for clinical translation. This technique has already been used in the autologous induced stem cell-derived retinal treatment for macular degeneration [30]. Moreover, minicircle vectors are also used as a method for cellular reprogramming and consist of minimal vectors containing only the eukaryotic promoter and the cDNA(s) that will be expressed. This technique was able to reprogram human adipose stromal cells, but the reprogramming efficiency is substantially lower (~0.005%) when compared to lentiviral-based techniques, for example [31].

HACs are also non-integrative systems for gene delivery with the main advantage of being able to transfer multiple genes and large sequences, which can be combined with sequences that increase therapy security and expression control. The authors constructed two different HACs, and the reprogramming of mouse embryonic fibroblasts into iPS cells was better achieved when the artificial chromosome also encoded a p53-knockdown cassette. The iPS cells were uniformly generated, and a built-in safeguard system was included, consisting of a reintroduced HAC encoding the Herpes Simplex virus thymidine kinase, which allowed the targeted elimination of reprogrammed cells by ganciclovir treatment [19].

Another promising strategy focusing on non-integrative non-viral reprogramming methods for iPS cell generation is through RNA molecules, such as micro-RNAs. These sequences are small endogenous non-coding RNAs that play important post-transcriptional regulatory roles [32]. They also repress gene expression through translational inhibition or by promoting the degradation of mRNAs [33]. One study showed that normal human hair follicles could be reprogramed into human iPS cells via doxycycline-inducible pTet-On-tTS vectors inserted by electroporation. These constructs contained pre-microRNA members of the miR-302 cluster, including pre-miR-302a, 302b, 302c, and 302d [34]. Although the reprogramming efficiency was not reported in this study, it is known that iPS cells induced by micro-RNAs have a reprogramming efficiency above 10% and also have the lowest tumorigenicity rate. Although this approach has not yet been used in any clinical test, it may help in future developments in regenerative medicine [33]. More recently, micro-RNAs were used in combination with other reprogramming methods to increase reprogramming efficiency [5].

Another promising transgene-free approach is the direct mRNA transfection of synthetic modified coding sequences of the “Yamanaka factors” (c-Myc, Klf4, Oct4, and Sox2). This is a non-integrating method that can reprogram multiple human cell types to pluripotency very efficiently, avoiding the antiviral immune response. The authors further showed that the same technology efficiently directed the differentiation of RNA-induced pluripotent stem cells (RiPSCs) into terminally differentiated myogenic cells [35]. The method of the direct delivery of synthetically transcribed mRNAs triggered somatic cell reprogramming with higher efficiency when compared to retroviruses [35]. These mRNAs are commercially available, and the authors used cationic lipid delivery vehicles for transfection in cell culture for seven days [27]. Similar alternatives are emerging as the cellular introduction of all reprogramming factors via a single synthetic polycistronic RNA replicon that requires single transfection [36]. In this case, the transfection of adult fibroblasts resulted in an efficient generation of iPS cells with the expression of all stem cell markers tested, consistent global gene expression profile, and in vivo pluripotency for all three germ layers.

Transgene-free cellular reprogramming has also been achieved using recombinant proteins. In this case, the generation of stable iPS cells was possible by directly delivering the four reprogramming proteins fused with a cell-penetrating peptide [22]. However, it has been technically challenging to synthesize large amounts of bioactive proteins that can cross the plasma membrane. This problem associated with low efficiency shows that much remains to be done for the use of recombinant proteins as a viable method. Two research groups were able to make enough bioactive proteins in an *E. coli* expression system and to reprogram mouse [37] and human fibroblasts [22]. More recently, Weltner and collaborators also used Clustered regularly interspaced short palindromic repeats (CRISPR)-associated Cas9 nuclease (CRISPR-Cas9)-based gene activation (CRISPRa) for reprogramming human skin fibroblasts into iPS cells [38]. CRISPR/Cas9 is a genome-editing tool powered by the design principle of the guide RNA that targets Cas9 to the desired DNA locus and by the high specificity and efficiency of CRISPR/Cas9-generated DNA breaks [39].

Another system for cellular reprogramming to generate iPS cells was the use of small-molecule compounds, which was developed by Hou and collaborators [23]. These authors used a combination of seven small molecules, but the efficiency achieved was only 0.2%. Small molecules have some advantages such as structural versatility, reasonable cost, easy handling, and no immune response. They can boost the application of iPS cells in disease therapy and drug screening. Some of these chemical compounds are valproic acid, trichostatin A (TSA), and 5-azacytidine, all capable of enhancing iPS cell generation [40]. One of the main advantages is that small (chemical) molecules can stimulate endogenous human cells to make tissue repair and regeneration in vivo, with no ectopic expression of factors. On the other hand, the method is time-consuming, and there is still a risk of genetic instability [6] to be overcome in future studies.

Despite all developments in the field of iPS cells, viral vector-based methods remain most popular among researchers [41]. Still, non-integrating non-viral self-excising vectors are more likely to be clinically applicable. To select an iPS cell reprogramming method, it is essential to maximize the capacity of cellular expansion in vitro, validate the detection and removal of incompletely differentiated cells, and search for genomic and epigenetic alterations. Probably, different somatic cell types will require different reprogramming methods to differentiate into the required terminal cell type in vivo.

Regardless of the reprogramming method, the risk of teratoma formation is inherent to iPS cells, as residual undifferentiated pluripotent cells retain very high plasticity. Although this risk has been reduced by highly sensitive methods for detecting remaining undifferentiated cells, teratoma formation cannot be ruled out [42]. Besides, c-Myc, one of the factors used for cellular reprogramming, is a well-known proto-oncogene, and its reactivation can give rise to transgene-driven tumor formation [43].

## 3. Applications of iPS Cells

IPS cells can differentiate into cells from any of the three primary germ layers [44], with great potential for clinical applications. Neurodegenerative disorders, for example, and diseases in which in vitro differentiation and transplant protocols have been established using conventional embryonic stem cells, are areas of immediate interest for iPS-based cell therapy. IPS cell lines can be generated in virtually unlimited numbers from patients affected by diseases of known or unknown causes. These cells can differentiate in vitro into the disease-affected cell type and offer an opportunity to gain insight into the disease mechanism to identify novel disease-specific drugs. In Table 2, we show examples of iPS cells generated from patients with sporadic or genetic diseases.

Some drugs that are in clinical trials were derived from iPS cell studies. For example, cardiomyocyte-derived iPS cells obtained from patients with type-2 long QT syndrome were used to test the efficacy and potency of new and existing drugs [51]. In regenerative medicine, iPS cells can be used for tissue repair or replacement of injured tissues after cell transplantation. Early trials using iPS cell transplantation focused on age-related macular degeneration, and this is a refractory ocular disease that causes severe deterioration in the central vision due to senescence in the retinal pigment epithelium (RPE). Preclinical studies showed good results in various animal models and corroborated the first clinical trial that began in 2014 [54]. Kamao and collaborators generated human iPS cells derived from RPE (hiPSC-RPE) cells that met clinical use requirements, including cellular quality and quantity, reproducibility, and safety. After the transplant, autologous non-human primate iPSC-RPE cell sheets showed no immune rejection or tumor formation [55]. Then, in the clinical trial using iPS cells, the cells were generated from skin fibroblasts obtained from patients with advanced neovascular age-related macular degeneration and were differentiated into RPE cells. In this test, autologous iPS cell-based therapy did not cause any significant adverse event [30]. However, the test with the second patient was discontinued due to genetic aberrations detected in the autologous iPS cells. With the rapid progress of genomic technologies, genetic aberrations in iPS cells will probably be reduced to a minimal level, with technological advances also focusing on automated closed culture systems [56].

Recent advances in genome editing technology have made it possible to repair genetic mutations in iPS cell lines derived from patients. Special attention has recently been focused on organoids derived from iPS cells, which are three-dimensional cellular structures mimicking part of the organization and functions of organs or tissues. Organoids were generated for various organs from both mouse and human stem cells, generating intestinal, renal, brain, and retinal structures, as well as liver organoid-like tissues, named liver buds [57]. Therefore, iPS cells-derived organoids can also be useful for drug testing and in vitro studies based on more complex cell models.

Moreover, iPS cells derived from cancer cells (cancer-iPS cells) can be a novel strategy for studying cancer. Primary cancer cells have been reprogrammed into iPS cells or at least to a pluripotent state, allowing the study and elucidation of some of the molecular mechanisms associated with cancer progression [58].

## 4. Pre-Clinical and Clinical Tests

The possibility of using iPS cells in the treatment of various diseases has brought hope regarding their potential to treat an increasing number of conditions. As iPS cells can be differentiated into all different cell types, new prospects for studying diseases and developing treatments by regenerative medicine and drug screening have emerged. Therefore, a large number of clinical and preclinical trials are being carried out [59] to treat human diseases using iPS cells.

The reprogramming of somatic cells was demonstrated using different animal species, including mouse, rat [60], dog [61], a variety of non-human primate species [62], pig [63], horse [64], cow [65], goat [66], and sheep [67]. However, once the goal of pre-clinical trials is the clinical use of iPS cells, a number of these trials are being conducted using human iPS cells. For specific applications, however, human cells are expected to be rejected by the animal hosts, and immunosuppressive protocols are required for long-term observation. On the other hand, immunomodulating drugs may affect the disease phenotype, and careful planning of every step is necessary. Any stem-cell-based clinical trial must follow all precedents already established for the evaluation of small biological molecules or human tissue remodeling and must be safe and effective. The production of cells must be carried out in facilities that follow the current Good Manufacturing Practices (GMP) and have stringent quality control for reagents with well-defined product release and potency assays. GMP is a set of conditions that define the principles and details of the manufacturing process, quality control, evaluations, and documentation for a particular product. Moreover, the best delivery system of iPS cells must be evaluated for each disease, which can be the use of intravascular catheters or surgical injection, for example.

Human-derived iPS cell lines successfully repopulated the murine cirrhotic liver tissue with hepatic cells at various differentiation stages. They also secreted human-specific liver proteins into mouse blood at concentrations comparable to those of proteins secreted by human primary hepatocytes [68]. In other preclinical studies, iPS cells were generated using adult dsRed mouse dermal fibroblasts via retroviral induction, following transplantation into the eye of immune-compromised retinal degenerative mice. After thirty-three days of differentiation, a large proportion of the cells expressed the retinal progenitor cell marker Pax6 and photoreceptor markers. Therefore, adult fibroblast-derived iPS cells are a viable source for the production of retinal precursors to be used for transplantation and treatment of retinal degenerative disease [69]. IPS cells were also generated from nonobese diabetic mouse embryonic fibroblasts or nonobese diabetic mouse pancreas-derived epithelial cells and differentiated into functional pancreatic beta cells. The differentiated cells expressed diverse pancreatic beta-cell markers and released insulin in response to glucose and KCl stimulation. Moreover, the engrafted cells responded to glucose levels by secreting insulin, thereby normalizing blood glucose levels, showing that these cells may be an important tool to help in the treatment of diabetic patients [70]. Human cardiomyocytes derived from iPS cells are another source of cells capable of inducing myocardial regeneration for the recovery of cardiac function. These cells were established using human dermal fibroblasts transfected with a retrovirus carrying the conventional factors Oct3/4, Sox2, Klf4, and c-Myc. When the iPS cells were transplanted into the myocardial infarcted area in a porcine model of ischemic cardiomyopathy, the activation of WNT signaling pathways induced cardiomyogenic differentiation. It was also observed that the transplanted cells significantly improved cardiac function and attenuated left ventricular remodeling [71]. In another study, dopaminergic neurons derived from protein-induced human iPS cells exhibited gene expression, physiology, and electrophysiological properties similar to the dopaminergic neurons found in the midbrain. The transplantation of these cells significantly rescued the motor deficits of rats with striatal lesions, an experimental model of Parkinson’s disease [72]. Moreover, after stroke-induced brain damage, adult human fibroblast-derived iPS cells were transplanted into the cortical lesion and, one week after the transplantation, there was the initial recovery of the forepaw movements. Moreover, engrafted cells exhibited electrophysiological properties of mature neurons and received synaptic input from host neurons [73].

In October 2018, 2.4 million iPS cells reprogrammed into dopaminergic precursor cell neurons were implanted into the brain of a patient in his 50s. In the three-hour procedure, the team deposited the cells into twelve sites, known to be centers of dopamine activity. The patient showed no significant adverse effects [74]. The first allogeneic clinical trial using iPS cells derived from mesenchymal stem cells for the treatment of graft-versus-host disease has also been reported, and no treatment-related serious adverse effects were observed [75]. Other clinical studies using iPS cells are being conducted in patients with heart failure [76,77]. Moreover, other tests have been approved for neural precursor cells for spinal cord injuries [78] and corneal epithelial cell sheets for corneal epithelial stem cell deficiency [79]. Thus, ongoing clinical tests provide a better understanding of clinical aspects involving immunosuppressants and fundamental elements such as genomic data that will pave the way for therapies using iPS cells.

### 4.1. Use of iPS Cells in Neurodegenerative Diseases

The iPS cells have the potential to revolutionize the field of neurodegenerative diseases, which are characterized by the progressive deterioration of neuronal function. Therefore, multiple capacities are affected, leading to cognitive impairment, memory deficits, deficiency in motor function, loss of sensitivity, dysfunction of the autonomous brain system, changes in perception, and mood [80]. Among neurodegenerative diseases, Alzheimer’s disease is the most prevalent form of dementia, characterized by the accumulation of amyloid-beta (Aβ) plaques and Tau-laden neurofibrillary tangles. Tau is a microtubule-associated protein found in the axons of the nerve cells, and these aggregates and tangles are the histopathological hallmarks of the disease [81]. The dysfunction and degeneration of neurons indeed underlie much of the observed decline in cognitive function, but various other types of non-neuronal cells are increasingly being implicated in the disease progression [82]. Therefore, iPS cells are emerging as an invaluable tool to better modeling the complex interactions that occur between multiple cell types in vivo. 3D and co-culture systems of iPS-derived cells in vitro hold promise to better understand the relevance of multiple cell types and the pathomechanisms that underlie the disease progression. Therefore, iPS cells have been generated from patients and healthy donors to study multiple genetic mutations in neurons, astrocytes, oligodendrocytes, microglia, pericytes, and vascular endothelial cells [83]. Moreover, a mutant Tau model derived from iPS cells was generated and showed several phenotypes associated with this neurodegenerative disease, including the pathogenic accumulation of Tau for drug screening [84]. Choi et al., 2014 showed a 3D culture model based on iPS cells that histopathologically reproduces the hallmarks of Alzheimer’s disease, including a robust extracellular deposition of Aβ. This model was sensitive to drugs, which reversed the pathological phenotype [85]. The use of neural models derived from iPS cells can validate molecular mechanisms identified in the disease models in rodents, for example, and play an important role in the discovery and screening of new drugs [86].

Parkinson’s disease is another important disease; being the second most common neurodegenerative disorder, it affects 2% to 3% of the population over 65 years of age. Characteristic features of Parkinson’s disease include neuronal loss in specific areas of the substantia nigra and widespread intracellular protein (α-synuclein) accumulation [87]. Due to the loss of dopaminergic neurons in localized regions of the brain, the use of human cells for therapeutic purposes has been studied with special attention. These assessments include iPS cells, whose good results supported the deployment of some studies that are already in the clinical phase. Pre-clinical studies have shown the efficient generation of iPS cells-derived dopaminergic motor neurons from non-human primates. Then, these cells were efficiently transplanted into a model of Parkinson’s disease in rats [88]. Several new protocols have improved the efficiency of obtaining dopaminergic neurons from iPS cells for the study and modeling of Parkinson’s disease [89]. The iPS cells used in some studies were mainly from patients carrying mutations in synuclein alpha, leucine-rich repeat kinase 2, PTEN-induced putative kinase 1, parkin RBR E3 ubiquitin-protein ligase, cytoplasmic protein sorting 35, and variants in glucosidase beta acid [90]. Although improvements are still needed, iPS cells make it possible to develop patient-specific disease models using disease-relevant cell types. Interestingly, using a human iPS cells-derived model of Parkinson’s disease, it was found that the myocyte enhancer factor 2C-peroxisome proliferator-activated receptor-γ coactivator-1α (MEF2C-PGC1α) pathway may be a new therapeutic target for Parkinson’s disease. The data from this study provided mechanistic insight into gene–environmental interaction in the pathogenesis of the disease [91]. Thus, it is important to develop models of neurodegenerative diseases using iPS cells because they involve a complex interplay of genetic alterations, transcriptional feedback, and endogenous control by transcription factors. Probably, the combination of different experimental approaches, using cellular systems and animal models, will increase the successful translation to the clinical practice [92].

In a successful pre-clinical study, the authors demonstrated that human dopaminergic neurons generated from iPS cells, and transplanted into a primate model of Parkinson’s disease, established connections with the host monkey brain cells with no tumor formation after two years [93]. Immediately after the successful animal experiments, the Japanese research group implanted ‘reprogrammed’ stem cells into the brain of a patient with Parkinson’s disease for the first time in 2018 (as NEWS Reported by Nature https://www.nature.com/articles/d41586-018-07407-9).

Recently, extracellular vesicles/exosomes derived from iPS cells of different lineages were involved in neurological diseases. Extracellular vesicles are lipid-enclosed structures with a diameter of 30–1000 nm, carrying transmembrane and cytosolic proteins. Exosomes are a subset of extracellular vesicles, with a diameter ranging between 30 and 200 nm. Functionally, they play an important role in intercellular communications, immune modulation, senescence, proliferation, and differentiation in various biological processes, and are vital in maintaining tissue homeostasis [94]. On the other hand, and as cited before, abnormal protein aggregation has been implicated in many neurodegenerative processes that lead to human neurological disorders. Recent reports suggested that exosomes combine these two important characteristics, as they are involved in the intercellular transfer of macromolecules, such as proteins and RNAs, and seem to play an important role in the aggregate transmission among neurons [95]. The authors showed that extracellular vesicles from iPS cells carry proteins and mRNA that can induce or maintain pluripotency, which can be used in regenerative strategies for neural tissue [96]. If this is true, extracellular vesicles/exosomes derived from corrected iPS cells, which do not accumulate protein aggregates, may be safer for human treatment than iPS cells themselves [94]. The infusion of neuronal exosomes into the brains of a murine model of Alzheimer’s disease decreased the Aβ peptide and amyloid depositions [97]. Moreover, exosomes obtained from stem cells were able to rescue dopaminergic neurons from apoptosis [98]. The authors showed that extracellular vesicles from mesenchymal stem cells, when injected into a mouse model of Alzheimer’s disease, reduced the Aβ plaque burden and the number of dystrophic neurites in the cortex and hippocampus [99]. Extracellular vesicles were also derived from human iPS neural stem cells and used for stroke treatment [100]. The results using extracellular vesicles/exosomes obtained from iPS cells point to a promising future in the treatment of neurodegenerative diseases.

### 4.2. Use of iPS Cells in Muscular Dystrophies

Muscular dystrophies (MD) are a group of genetic diseases that lead to skeletal muscle wasting and may affect many organs (multisystem) [101]. The terminal pathology often shows muscle fibers necrosis and muscle tissue replacement by fibrotic or adipose tissues. Currently, there is no cure for MD, and the available treatments are palliative or of limited effectiveness [102]. The most frequent and one of the most severe forms of all MD is the Duchenne muscular dystrophy (DMD), a muscle pathology caused by the lack of the protein dystrophin. In this case, previous cell-based therapies did not show satisfactory results after myoblast transplantation [103]. Myoblasts are the progeny of muscle satellite cells (SC), the main stem cell population found in adult skeletal muscles. Quiescent SCs are triggered to reenter into the cell cycle mainly by muscle damage, and the SC-derived myoblasts proliferate and fuse to form new multinucleated myofibers [101]. In most myoblast-based therapies, allogeneic cells were obtained from muscle biopsies from healthy donors, resulting in transplanted cell rejection by the immune system, with low surviving rates, poor dispersion, and differentiation [103,104,105]. With the advances of iPS cell technology, some of these issues are being addressed (Figure 2).

One of the main problems in the application of stem cell therapy in muscle diseases is to obtain large numbers of cells for sufficient engraftment, and the use of iPS cells may overcome this obstacle. For this purpose, Darabi and colleagues [106] applied the conditional expression of Pax7 to iPS cells, a transcription factor that plays a role in SC proliferation. Then, Pax7^+^ iPS cells were obtained on a larger scale for transplant into a mouse dystrophic muscle, which showed dystrophin^+^ fibers with superior strength [106]. Moreover, the authors genetically restored the dystrophin expression in autologous iPS cells derived from DMD patients. For this, three corrective methods were used, which were exon knock-in, exon skipping, and frameshifting, and the exon knock-in was the most effective approach [107]. The Cas9 protein (CRISPR-associated protein 9), derived from type II CRISPR (clustered regularly interspaced short palindromic repeats) bacterial immune systems, is a technology that has also emerged as an approach capable of targeting the mutated dystrophin gene, aiming to rescue its expression in vitro in iPS cells derived from selected patients [108].

Moreover, using CRISPR-Cas9 technology with single guide RNA, dystrophin expression was restored by exon skipping through myoediting in iPS cells. The genetic alterations observed in the multiple patients included large deletions, point mutations, or duplications within the *DMD* gene. The corrected iPS cells efficiently restored the expression of dystrophin and the corresponding mechanical contraction force in derived cardiomyocytes [109]. In summary, several methods of gene editing have been applied for the correction of the *DMD* gene to allow the transplantation of genetically corrected autologous iPS cells. Of these, the CRISPR-Cas9 system, in particular, has passed multiple proof-of-principle tests and is now being used in pre-clinical trials (Figure 2).

Reprogrammed fibroblast- and myoblast-derived iPS cells were also obtained from patients with limb-girdle muscular dystrophy type 2D (LGMD2D). This disease is a sarcoglycanopathy caused by mutations in the *SCGA* gene, which provides instructions for making the alpha component of the sarcoglycan protein complex. This multiprotein complex plays a role in the anchoring of the dystrophin-glycoprotein complex (DGC) to the extracellular matrix and helps to maintain muscle fiber membrane integrity. The iPS cells were expanded and genetically corrected in vitro with a lentiviral vector carrying the human gene encoding the α-sarcoglycan. Finally, the transplantation of mouse iPS cells into α-sarcoglycan-null immunodeficient mice, an experimental model of the disease, resulted in the amelioration of the dystrophic phenotype [110]. This transplant also showed that iPS cells restored the compartment of SC, an essential checkpoint for sustained muscle regeneration.

Recently, Perepelina and collaborators generated iPS cells from Emery–Dreifuss muscle dystrophy associated with the genetic variant LMNAp.Arg527Pro. Patient-specific peripheral blood mononuclear cells were reprogrammed using the Sendai virus system, and the authors comment that this is a useful tool to study laminopathies in vitro [111]. Moreover, using three-dimensional (3D) tissue engineering techniques, artificial skeletal muscle tissue was generated using iPS cells from patients with Duchenne, limb-girdle, or congenital muscular dystrophies [112]. In this way, artificial muscles recapitulated characteristics of human skeletal muscle tissue, providing an invaluable tool to study pathological mechanisms, drug testing, cell therapy, and the development of tissue replacement protocols.

## 5. Conclusions

The use of iPS cells still has many challenges ahead before they can be clinically used in the supportive treatment of patients with MD. Among these, we can cite the injection of iPS cells (or muscle-committed iPS-derived cells) into large muscles, the immunological recognition of proteins expressed only after the genetic correction, the capacity of cellular dispersion through the muscle, the number of therapeutic interventions needed to replenish cellular muscle populations, the ability to produce corrected SC for sustained muscle recovery, and the control of transplanted cells death.

To address these and other limitations, we propose that autologous iPS cells be submitted to multiple treatments aiming to improve cellular engraftment and clinical use. Besides the genetic correction of underlying pathological mutations, these cells can be further treated in culture to boost cell proliferation, long-term survival, dispersion in the muscle, differentiation into muscle fibers, and others. We proposed before the use of multiple combined in vitro treatments for adoptively transferred myoblasts for cell-based therapy, and these are summarized in [101]. These treatments include vascular endothelial growth factor (VEGF), insulin-like growth factor-1 (IGF-1) and basic fibroblast growth factor (bFGF), Wnt7a, Ursolic acid, and extracellular matrix components. Moreover, the recipient muscle to be injected with the corrected and boosted iPS cells can also be treated to favor the engraftment. These treatments include actinin receptor type 2B inhibitor, IL-6, JAK/STAT 3 inhibitor, growth factors, the coinjection of other supportive cell types, such as macrophages and fibroblasts, and others.

We believe that the correct choice for the ideal combination of the cell type to be reprogrammed into iPS cells, the technical procedure for genetic correction, the in vitro treatments to boost iPS cells, and the in vivo preparation of recipient’s muscles, hold the key for a more successful application of iPS cells in clinical translation. However, we believe that systemic treatments consisting of the injection of cells will not lead to individual muscle damage and strength improvement. The transplanted cells do not express the required repertoire of molecules necessary for endothelial transmigration. Probably, selected individual and more affected muscles are more likely to benefit from cellular-based therapies, followed by treatments that can increase injected cell dispersion within the muscle.

Currently, public–private partnership consortia are providing resources to form iPS cell banks for clinical and research purposes. These banks have coordinated standards to meet international criteria for quality-controlled repositories of iPS cells. Although the use of iPS cells for autologous therapy seems more appropriate, having allogeneic banks of iPS cells already generated and tested would reduce the time needed to start treatment, decrease costs, and increase the chances of recovery of treated individuals [113]. Thus, although many technical challenges must still be overcome, the technology of iPS cells has already taken a marked leap in clinical management and in vitro models to study and treat diseases.

## Figures and Tables

**Figure 1 ijms-21-05467-f001:**
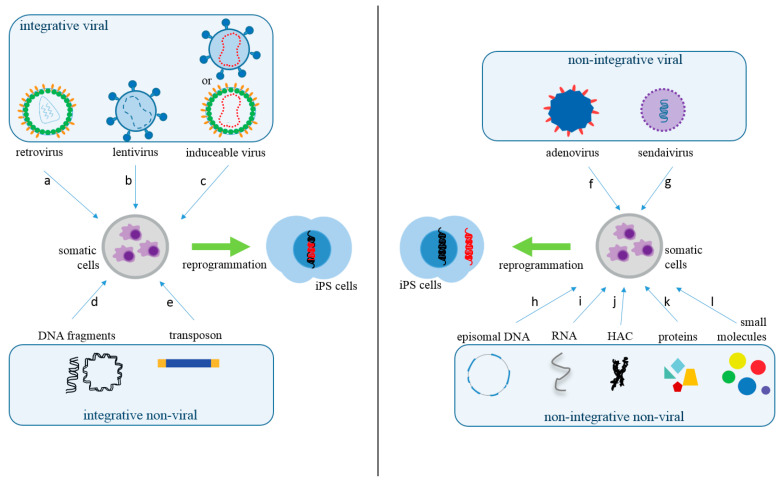
Somatic Cells Reprogramming Methods. The methods used to produce iPS cells can be classified into integrative viral, such as retrovirus (a), lentivirus (b), or inducible retro or lentivirus (c); and integrative non-viral, such as linear or circular DNA fragments (d) or transposons (e). In regards to non-integrative methods, they can also be separated as viral, such as adenovirus (f) or Sendai virus (g). Non-integrating non-viral methods are episomal DNA (h), RNAs (i), human artificial chromosome (HAC) (j), proteins (k), or small molecules (compounds) (l). The red DNA represents epigenetic inserted sequences for cellular reprogramming.

**Figure 2 ijms-21-05467-f002:**
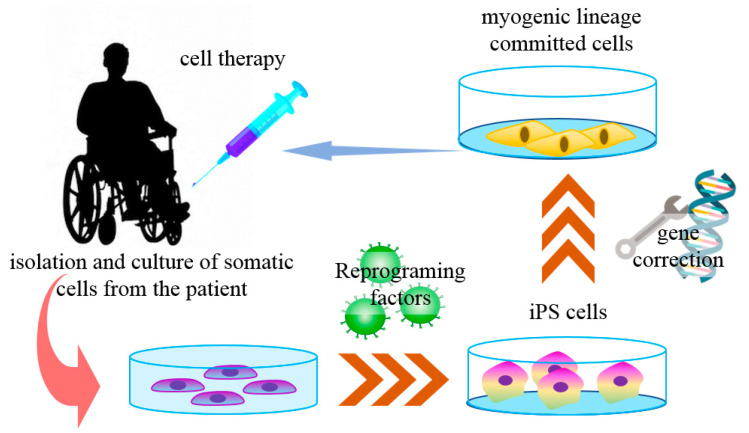
iPS cells in Duchene muscular dystrophy cell therapy. The somatic cells derived from specific patients with Duchenne muscular dystrophy (DMD) can be reprogrammed into iPS cells with reprogramming factors. These cells are then genetically corrected to express the protein dystrophin for the autologous muscular injection of muscle-committed cells.

**Table 1 ijms-21-05467-t001:** Comparison of multiple reprogramming techniques.

Vector Technology	Introduced Factors	Advantages	Disadvantages	Ref.
retrovirus	OCT3/4, SOX2, KLF4, c-MYC, NANOG	technically easy, reasonable efficiency, low costs	low safety, with risk tumorigenesis	[4]
lentivirus	OCT3/4, SOX2, KLF4, cMYC, UTF1, p53, siRNA, Slc7a1	higher efficiency than a retrovirus	low safety, with risk tumorigenesis	[12]
adenovirus	OCT3/4, SOX2, KLF4, c-MYC	transient gene expression	very low efficiency	[13]
Sendai virus	OCT3/4, SOX2, KLF4, c-MYC	higher efficiency than a retrovirus	expensive kits	[14]
piggyBac transposon	OCT3/4, SOX2, KLF4, c-MYC	safe and with a precise deletion	low efficiency	[15]
plasmid DNA	OCT3/4, SOX2, KLF4, L-MYC, LIN28, p53 shRNA	slightly higher average safety level	low efficiency	[16]
episomal DNA	OCT4, SOX2, NANOG, KLF4, c-MYC, LIN28, SV40LT	satisfactorily safe	low efficiency	[17]
minicircle DNA	OCT4, SOX2, LIN28, NANOG	easy to handle, safe	low efficiency when compared to viral methods	[18]
human artificial chromosome	OCT/4, SOX2, KLF4, c-MYC, p53 shRNA	built-in safeguard system	low efficiency and time-consuming	[19]
microRNA	miR-200c, miR-302 s, miR-369 s family miRNAs	proper safety	less efficient than mRNA, time-consuming, fast microRNA degradation	[20]
mRNA	OCT4, SOX2, KLF4, c-MYC, LIN28	proper safety, high efficiency	Multiple rounds of transfection are required	[21]
protein	OCT3/4, SOX2, KLF4, c-MYC	proper safety	Very low efficiency, requires large quantities of pure proteins	[22]
Small molecules	HIR, 616452, FSK, DZNep, PD0325901, VPA, Tranylcypromine, TTNPB	proper safety, easy to handle	low efficiency, time-consuming	[23]

**Table 2 ijms-21-05467-t002:** Examples of terminally differentiated cells generated from induced pluripotent stem (iPS) cells.

Disease	Differentiated Cell Type	Reference
Parkinson’s disease	dopaminergic neurons	[45]
Huntington’s disease	ND *	[46]
amyotrophic lateral sclerosis	motor neurons	[47]
spinal muscular atrophy	motor neurons	[48]
Fanconi anemia	blood cells	[49]
LEOPARD syndrome	cardiomyocytes	[50]
congenital long QT syndrome	cardiomyocytes	[51]
Duchenne muscular dystrophy	ND	[46]
type I diabetes	beta cells	[52]
alpha1-antitrypsin deficiency	hepatocytes	[53]
familial hypercholesterolemia	hepatocytes	[53]
Down syndrome	ND	[46]

* ND means not done.

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
