# Peer review of "Induced Pluripotent Stem Cells: Hope in the Treatment of Diseases, including Muscular Dystrophies"

_ijms, 2020, doi:10.3390/ijms21155467_

Round 1
Reviewer 1 Report
The reviewer has several comments to be addressed. 1) The authors may want to introduce Table 1 earlier in the chapter 2 for readers to better understand this paper. 2) Could the authors discuss the delivery system of iPS cells to the whole body muscle tissues? The reviewer wonders if another major bottleneck of iPS therapy for MD would be how to deliver iPS cells to the diseased muscle tissues. 3) Line 305-308: Could the authors provide information on therapeutic effects of iPS therapy for Parkinson’s disease? 4) Line 344: The authors may want to spell out and explain for CRISPR-Cas9 technology. 5) References: #1 and #4 are duplicated.
Author Response
Dear editor and reviewers
We are deeply thankful for the invaluable suggestions and contributions that were made to the manuscript. Please, find below a point-by-point rebuttal addressing each comment.
Reviewer 1:
The reviewer has several comments to be addressed.
1) The authors may want to introduce Table 1 earlier in the chapter 2 for readers to better understand this paper.
It was done. We transferred the Table to the beginning of the topic; it is now close to Figure 1. We also formatted the appearance of the Table.
2) Could the authors discuss the delivery system of iPS cells to the whole body muscle tissues? The reviewer wonders if another major bottleneck of iPS therapy for MD would be how to deliver iPS cells to the diseased muscle tissues.
It is true; we did not mention clearly what our position was about the treatment via, either systemic or in selected and individual muscles. According to vast literature in the field, we believe that circulating injected cells do not express the required repertoire of molecules necessary for endothelial transmigration. In this case, most cells die or are found in the liver or spleen. We believe that more affected and selected muscles should be subjected to cellular injection, and additional treatments (commented in the Conclusion topic) may improve cellular dispersion within the treated muscle. This commentary was included in line 485, as follows (also marked in yellow):
“However, we believe that systemic treatments consisting of the injection of cells, will not lead to individual muscle damage and strength improvement. The transplanted cells do not express the required repertoire of molecules necessary for endothelial transmigration. Probably, selected individual and more affected muscles are more likely to benefit from cellular-based therapies, followed by treatments that can increase injected cell dispersion within the muscle.”
3) Line 305-308: Could the authors provide information on the therapeutic effects of iPS therapy for Parkinson’s disease?
This was a valuable commentary, and we included a whole topic named: “4.1. Use of iPS cells in neurodegenerative diseases”. We do believe that the manuscript is now more complete and will serve to a broader audience.
4) Line 344: The authors may want to spell out and explain for CRISPR-Cas9 technology.
It was done, it is marked in yellow: line 430
5) References: #1 and #4 are duplicated.
It was corrected. However, this section formatting was lost.
Reviewer 2 Report
The manuscript of Gois Berghini et al. is a review of literature about the methods of cell reprogramming and the results of clinical studies on several diseases. The Author salso discucc the possibility of applying iPS cells in the treatment of muscular dystrophies.
In my opinion, the review could be of interest and suitable for publication on IJMS. On the other hand, the review may be even more attractive with the inclusion of the possibility of using iPS-derived exosomes in the treatment of degenerative diseases considered in the manuscript.
Author Response
Dear editor and reviewers
We are deeply thankful for the invaluable suggestions and contributions that were made to the manuscript. Please, find below a point-by-point rebuttal addressing each comment.
reviewer 2:
The manuscript of Gois Berghini et al. is a review of literature about the methods of cell reprogramming and the results of clinical studies on several diseases. The Author salso discucc the possibility of applying iPS cells in the treatment of muscular dystrophies.
In my opinion, the review could be of interest and suitable for publication on IJMS. On the other hand, the review may be even more attractive with the inclusion of the possibility of using iPS-derived exosomes in the treatment of degenerative diseases considered in the manuscript
Thank you for this pertinent suggestion, a whole topic was inserted, as follows: “4.1. Use of iPS cells in neurodegenerative diseases”.
Kind regards
Dr. Andrea Henriques-Pons